# Dual-Modeling Decouple Distillation for Unsupervised Anomaly Detection

Xinyue Liu
School of Computer Science and Engineering
Beihang University
Beijing, China
liuxinyue7@buaa.edu.cn

Jianyuan Wang*
Key Laboratory of Intelligent Bionic Unmanned Systems,
Ministry of Education
School of Intelligence Science and Technology
University of Science and Technology Beijing
Beijing, China
wangjianyuan@ustb.edu.cn

Biao Leng
School of Computer Science and Engineering
Beihang University
Beijing, China
lengbiao@buaa.edu.cn

Shuo Zhang
Beijing Key Lab of Traffic Data Analysis and Mining,
School of Computer & Technology
Beijing Jiaotong University
Beijing, China
zhangshuo@bjtu.edu.cn

## Abstract

Knowledge distillation based on student-teacher network is one of the mainstream solution paradigms for the challenging unsupervised Anomaly Detection task, utilizing the difference in representation capabilities of the teacher and student networks to implement anomaly localization. However, over-generalization of the student network to the teacher network may lead to negligible differences in representation capabilities of anomaly, thus affecting the detection effectiveness. Existing methods address the possible over-generalization by using differentiated students and teachers from the structural perspective or explicitly expanding distilled information from the content perspective, which inevitably results in an increased likelihood of underfitting of the student network and poor anomaly detection capabilities in anomaly center or edge. In this paper, we propose **D**ual-**M**odeling **D**ecouple **D**istillation (**DMDD**) for the unsupervised Anomaly Detection. In DMDD, a Decouple Student-Teacher Network is proposed to decouple the initial student features into normality and abnormality features. We further introduce Dual-Modeling Distillation based on normal-anomalous image pairs, fitting normality features of anomalous image and the teacher features of the corresponding normal image, widening the distance between abnormality features and the teacher features in anomalous regions. Synthesizing these two distillation ideas, we achieve anomaly detection which focuses on both edge and center of anomaly. Finally, a Multi-perception Segmentation Network is proposed to achieve focused anomaly map

fusion based on multiple attention. Experimental results on MVTec AD show that DMDD surpasses SOTA localization performance of previous knowledge distillation-based methods, reaching 98.85% on pixel-level AUC and 96.13% on PRO.

## CCS Concepts

• **Computing methodologies** → **Visual inspection**; **Image segmentation**; *Hierarchical representations*.

## Keywords

Anomaly detection, Knowledge distillation, Unsupervised learning

**ACM Reference Format:**
Xinyue Liu, Jianyuan Wang, Biao Leng, and Shuo Zhang. 2024. Dual-Modeling Decouple Distillation for Unsupervised Anomaly Detection. In *Proceedings of the 32nd ACM International Conference on Multimedia (MM '24), October 28-November 1, 2024, Melbourne, VIC, Australia.* ACM, New York, NY, USA, 10 pages. https://doi.org/10.1145/3664647.3681669

## 1 Introduction

Anomaly Detection (AD) is an important task in the field of computer vision, aiming to detect and locate anomalous regions in images, which has wide applications such as industrial quality inspection [3, 4], medical disease screening [35, 43], and video surveillance [19, 22, 29, 36]. Considering the scarcity of anomalous samples, AD is usually unsupervised, relying only on normal samples to train the model.

Most existing unsupervised AD methods rely on the difference in the feature distribution or reconstruction ability of the network for normal pixels and anomalous pixels to locate anomalies in the image. Inspired by this idea, Knowledge Distillation (KD) based on Student-Teacher (S-T) Network gradually becomes one of the mainstream paradigms of unsupervised AD in recent years. In the training process, the student network learns the feature representation of normal images by the pre-trained teacher network, so that only the normality representation ability is obtained. During inference, the features extracted by the teacher network are used as comparison

---

*Corresponding author.

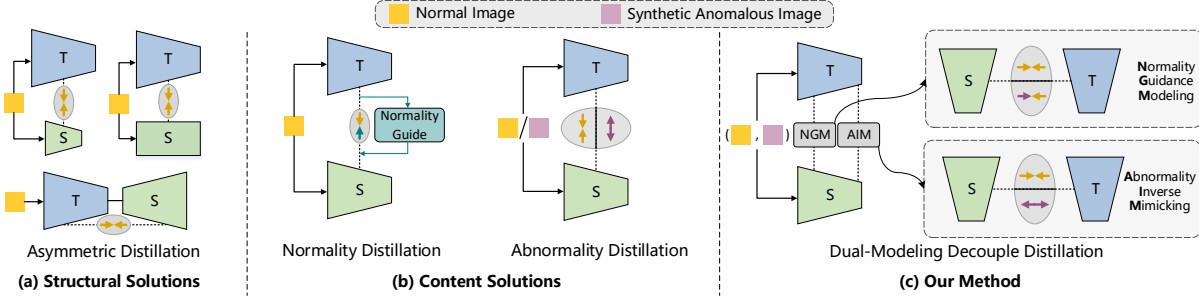

**Figure 1: (a)** *Structural Solutions.* **(b)** *Content Solutions.* **(c)** Our proposed *Dual-Modeling Decouple Distillation* method with dual-branch carrying out distillation with different concerns.

benchmarks. The similarity between the student features and the teacher benchmarks indicates anomaly localization results. That is, regions with larger feature similarity are more likely to be normal, while regions with small similarity are more likely to be anomalous.

These KD-based methods assume that the student network cannot obtain the representation ability of the teacher network for anomalous pixels in the images during training. However, due to the generalization ability of neural networks, there is possibility that the student network may learn to generate abnormality representations in practice, thus reversing the above assumption. Existing KD-based methods work on the over-generalization problem from both **structural** and **content** perspectives, as shown in Fig. 1.

Structurally, Asymmetric Distillation [11, 26, 27] is proposed, whose main idea is to take advantage of asymmetry to differentiate the information capacity of the teacher and student networks and prevent over-generalization. However, *the different capacities and network information transmission directions are likely to cause the student network to underfit the teacher network in terms of normality features* [11, 15]. As for content, some methods improve the basic S-T network in respect to the distillation of information from normality and abnormality. Normality Distillation [13] focuses on "normality forgetting" issue, and indirectly enlarges the feature differences in anomalous regions by guiding the student's normality feature generation using the teacher features through memory banks and other means. Abnormality Distillation [46] introduces anomaly synthesis into KD paradigm, and explicitly distances the student's features from the teacher's ones in the anomalous regions. However, for anomaly centers, anomalous pixels in the receptive field are not conducive to normality feature generation guidance; for anomaly edges, interference with normal pixels in the receptive field is not favorable for abnormality feature differentiation. Consequently, *improving distillation content only from normality or abnormality has limitation in capturing the full scope of anomalies.*

To tackle the above problems, in structure, our intuition is to construct a decoupled S-T network with the same capacity of teacher and student networks. In content, we propose to implement a dual-branch design by combining the ideas of Normality Distillation and Abnormality Distillation. Through fully aligned teacher and student networks, the adequacy of the representation ability of the student network is guaranteed. The design of dual-branch decoupling of the student network in turn ensures the difference between the student network and the teacher network. Moreover, with the feature

decoupling idea, the ideas of normality and abnormality modeling are able to be introduced into the same distillation framework. In this way, anomaly detection is synthetically realized from two aspects of fine (edge localization) and coarseness (center localization), overcoming problems of previous content solutions.

Following the above ideas, as in Fig. 1, we propose a Dual-Modeling Decouple Distillation framework to implement dual-branch coarseness-fine distillation. Our method mainly includes three components: Decoupled Student-Teacher Network, Dual-Modeling Distillation and Multi-perception Segmentation Network. First, letting the same pre-trained network be the student and teacher networks, we propose a Decoupled Student-Teacher Network, which aims to decouple normality feature and abnormality feature from the student output for subsequent distillation. In addition, a Dual-Modeling Distillation is innovatively proposed, which relies on normal-anomalous image pairs and contains two distillation modules: Normality Guidance Modeling and Abnormality Inverse Mimicking to remodel normality features and abnormality features in different directions. Finally, after obtaining the anomaly maps calculated based on the student's decoupled features and the teacher's features, a Multi-perception Segmentation Network is put forward for achieving precise fusion of the anomaly maps in a differentiated manner. We conduct unsupervised AD experiments on multiple datasets, and the results show that our method achieves SOTA performance compared with existing KD-based methods.

## 2  Related Works

Unsupervised image Anomaly Detection has been rapidly developed due to the difficulty in obtaining anomalous images. The image reconstruction-based methods are widely accepted for Anomaly Detection using autoencoders [6, 30], variational autoencoders [30, 48], generative adversarial networks [9, 23, 28] and diffusion models [21, 33, 34, 41, 44]. Memory-based methods [1, 24, 37] are also commonly employed for unsupervised AD by designing memory bank to store normal features during training. In addition, memory bank is often utilized as a mean of guiding normality feature generation in other paradigms [13, 15]. Besides, some methods use parametric density estimation [10, 16] to determine anomalous outliers, and by introducing the idea of Normalizing Flow [14, 38, 47], the detection effectiveness of this type of methods has reached a high level. Recently, some other methods [18, 40] propose anomaly synthesis

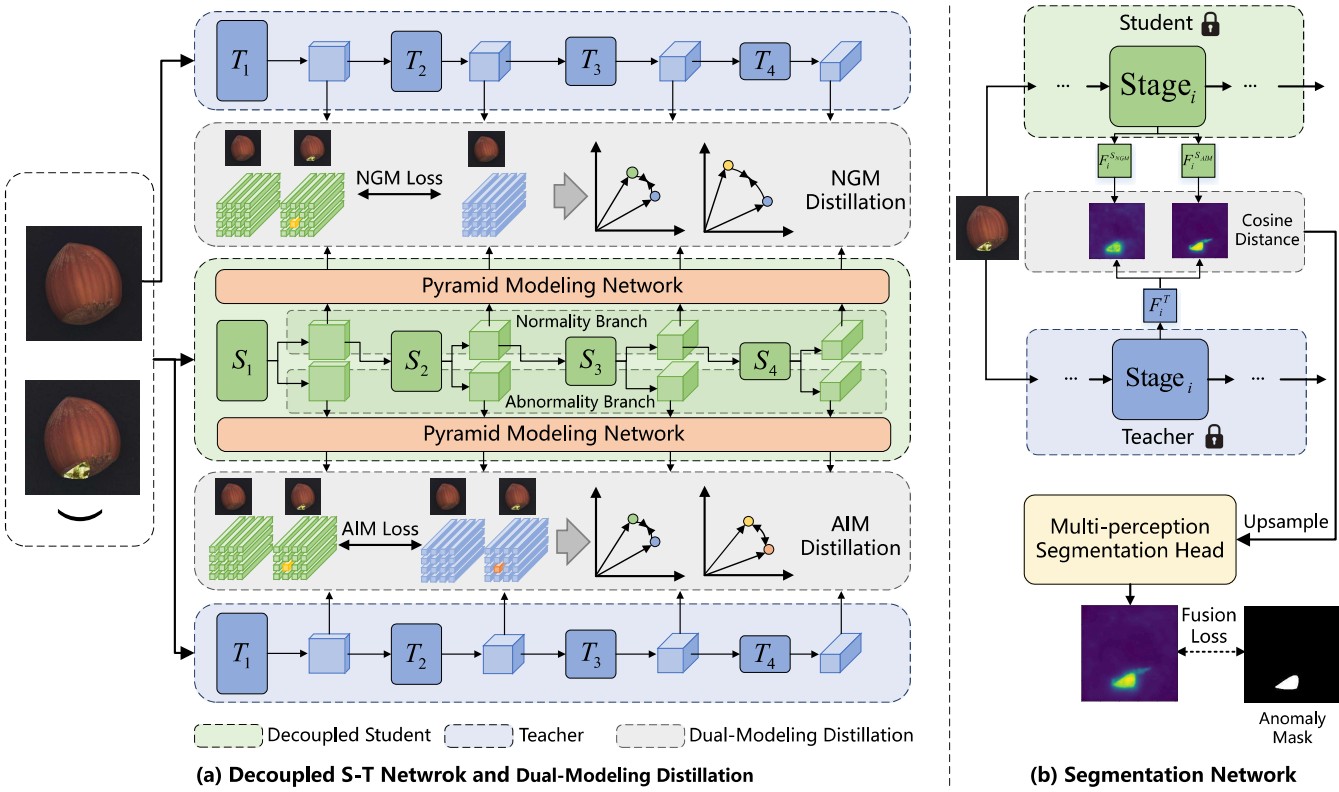

**(a) Decoupled S-T Netwrok and Dual-Modeling Distillation**

**(b) Segmentation Network**

**Figure 2: Overview of DMDD.** *Left*: Our proposed Decoupled Student-Teacher Network is demonstrated. First, the student network uses a dual-branch design to decouple normality features and abnormality features. Then, the decoupled features are distilled through Normality Guidance Modeling (NGM) and Abnormality Inverse Mimicking (AIM) respectively. *Right*: The Segmentation Network is shown, where a Multi-perception Segmentation Head is trained by the ground-truth masks of synthetic anomalies. The anomaly maps and anomaly scores are obtained directly during inference.

to transform the unsupervised anomaly detection into a supervised problem, which greatly improves the detection performance.

In addition, in recent years, knowledge distillation has gradually become a mainstream solution paradigm for unsupervised Anomaly Detection. The hypothesis is that a student network trained using teacher features on normal samples only obtains the teacher's representation of normality, but cannot simulate the teacher's abnormal representation. US [5] first introduces knowledge distillation into unsupervised AD. MKD [27] and STPM [32] use multi-scale features, where differentiated teacher and student network structures are also proposed to solve the over-generalization problem of the student network. Similarly, using the idea of differential teacher and student structures [2, 26, 45], RD [11] and RD++ [31] design reverse distillation with an encoder and a decoder as teacher and student respectively. There are other methods [7, 42] addressing over-generalization in terms of distillation content by introducing information such as synthetic anomalies [46], memory banks [13], and so on, to ensure that the student network generates different features in the anomalous regions from the teacher network.

## 3 Method

During the training process of unsupervised AD, there are only normal images $I_{train}^n = \{I_1^n, I_2^n, ..., I_k^n\}$ input into the model, which means the model is unable to obtain abnormality perception ability by explicitly training on anomalous images. However, the testing set $I^{test} = \{I_1^{test}, I_2^{test}, ..., I_s^{test}\}$ contains both normal images and anomalous images unseen during training. As a result, the training goal of the unsupervised AD model is to get the ability to detect and localize anomalous regions during the inference process.

### 3.1 Overall Framework

Based on the student-teacher framework of KD, we propose Dual-Modeling Decouple Distillation (DMDD) for unsupervised AD. The left part of Fig. 2 shows the proposed Decoupled Student-Teacher Network (Sec 3.2), which is the basis of distillation. Take the first 4 stages of WideResNet50 [39] pre-trained on ImageNet [12] as the teacher and student network. The two teacher networks share the same weights. During the training process, the input of the whole network is a normal-anomalous image pair $(I^n, I_a^n)$ containing a normal image $I^n$ and a synthetic anomalous image $I_a^n$. The weights of the teacher network are fixed, and the student network is continuously optimized by Dual-Modeling Distillation (Sec 3.3). In addition,

at the end of each epoch, the teacher and student are frozen and Multi-perception Segmentation Network (Sec 3.4) is optimized, as shown in the right part of Fig. 2. For inference, the anomaly map and anomaly score output by Multi-perception Segmentation Network are used for anomaly detection and localization.

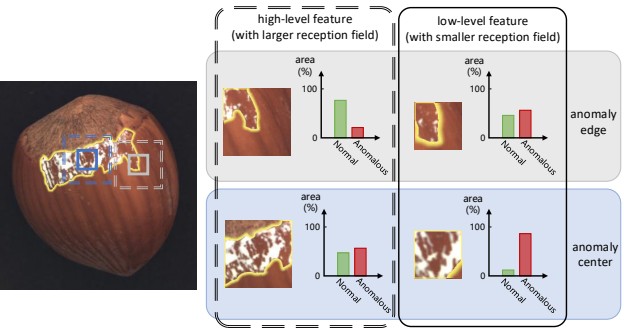

**Figure 3: The proportions of normal and anomalous pixels in receptive fields located at anomaly's center and edge.**

## 3.2 Decoupled Student-Teacher Network

Due to the size limitation of the receptive field of Convolutional Neural Network (CNN), distillation methods based on normality construction (Normality Distillation) often have stronger perception of the edge pixels of anomalous regions, and distillation methods based on synthetic anomalies (Abnormality Distillation) often have stronger perception of the center pixels of anomalous regions. This is because the receptive field at the edge of the anomalous region receives more normality information, which enables better reconstruction of normality. On the contrary, the convolution kernel receives more abnormality information in the center of the anomaly region, which prevents interference from normal pixels, and thus being able to better expand the distance between the student features and the teacher features. Fig. 3 illustrates the above reasoning with a diagram. Existing methods only use one of the above methods, leading to insufficient perception of the anomalies' center or edge, resulting in inaccurate anomaly localization results.

To solve this problem, we propose to decouple and remodel the features of student network to perceive anomalies from both the edge and center perspectives. Our idea is that the features of the student network corresponding to the anomalous region are able to be decomposed into the normality feature and abnormality feature. Among them, normality feature refers to the feature generated by the teacher network assuming that this region is normal, and abnormality feature is hoped to be different from the anomalous features of the teacher network.

Therefore, we carry out the dual-branch design including Normality Branch and Abnormality Branch for the student network and propose a Decouple Student-Teacher Network. In addition to providing decoupling features for the subsequent distillation process, this design also differentiates the teacher and student architectures to some extent, and structurally avoids over-generalization of the student network. We design to add a $1 \times 1$ convolution layer after each stage of the student network to expand the channel by 2 times,

and divide the output features into two parts along the channel dimension, which are used as initial normality feature and abnormality feature. Among them, the normality feature are used as the input of the next stage.

For deep CNNs, as in Fig. 3, the receptive field of high-level features is larger, while the receptive field of low-level features is smaller. That is, high-level features are greatly affected by surrounding pixels, while low-level features are less affected. For Normality Branch, the goal is to generate normality features on anomalous pixels. Therefore, considering that the larger the receptive field corresponding to the anomalous pixel is, the more normal information it contains, we believe that the normality features of higher stages are better generated. On the contrary, it is necessary for Abnormality Branch to generate features on anomalous pixels that are as different as possible from the teacher features. When the receptive field is large, the normal information that may appear in the receptive field corresponding to the anomalous pixel is likely to have an impact on the remodeling of the abnormality features. As a result, the normal information is more unlikely to be contained in the smaller receptive field, which means the abnormality feature remodeling at the lower stages is better.

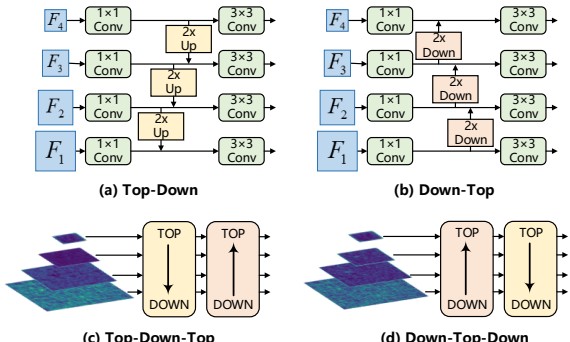

**Figure 4: Pyramid Modeling Network.**

Based on the analysis above, we introduce multi-scale feature modeling into the student network, and propose a bidirectional dual-path **Pyramid Modeling Network**, as shown in Fig. 4. For Normality Branch, a top-down-top pyramid feature modeling path is introduced. And a down-top-down feature modeling path is designed for Abnormality Branch. On one hand, Pyramid Modeling Network uses the better decoupled features to guide features with poor decoupling capabilities (inner path), and on the other hand, it increases the information contained in features by adding an opposite feature fusion path (outer path).

## 3.3 Dual-Modeling Distillation

By designing Normality Branch and Abnormality Branch, we get the initially decoupled normality features and abnormality features from the student network. To further utilize the features output by the teacher network to optimize the normality and abnormality features of the student network in the specific directions, we design Dual-Modeling Distillation.

The entire distillation framework is divided into two modules: Normality Guidance Modeling (NGM) and Abnormality Inverse

Mimicking (AIM), which remodel the normality features and abnormality features from two aspects. The features optimized by NGM and AIM remain similar to the teacher's features in normal regions, but are different from the teacher's features in anomalous regions.

### 3.3.1 Normality Guidance Modeling.
Since anomalous regions usually have different pixel distributions from normal regions, the features extracted by CNN in normal regions and anomalous regions are obviously different. Based on this experience, some methods use the similarity of the features between pixels or patches to determine whether there occurs anomaly. The key to this type of method is how to remember the normality features of the images.

Most of the previous methods [13, 15, 45] use memory bank or generative network to store or simulate normality features. However, memory bank requires a large storage space, and the operation of searching the memory bank has greater computational complexity. The generative network requires both a encoder and a decoder, which requires a large amount of calculation to generate normality features. In addition, designing the generative network as the student leads to a large structural difference between the teacher network, facing the risk of underfitting. Therefore, for the task of modeling normality features, we draw on the idea of Mask Image Modeling (MIM) used in self-supervised learning. Based on the synthetic anomalies as the masks and the normality features generated by the teacher network as the regression goal, we propose a novel distillation method Normality Guidance Modeling.

During training, input each normal image $I^n$ to the teacher and student networks at the same time. In addition, the input of the student network also includes the corresponding synthetic anomalous image $I_a^n$. NGM uses the feature $F_n^T$ corresponding to $I^n$ extracted by the teacher as distillation indicator, and minimizes the distances from the features generated by the student on $I^n$ and $I_a^n$ to $F_n^T$.

Let $F_{n\ i}^{SNGM}$ and $F_{a\ i}^{SNGM}$ represent the output normality features of the $i$-th stage of the student network for a pair of normal and synthetic anomalous images respectively. Let $F_{n\ i}^T$ represent the feature of the $i$-th stage corresponding to the normal image output by the teacher network. Then the loss $L_{NGM}$ used to train NGM distillation is calculated based on cosine similarity as

$$D_i^{n/a}(h, w) = 1 - \frac{F_{n\ i}^T(h, w)F_{n/a\ i}^{SNGM}(h, w)}{\|F_{n\ i}^T(h, w)\|\|F_{n/a\ i}^{SNGM}(h, w)\|} \quad (1)$$

$$L_{NGM} = \frac{1}{4}\frac{1}{H_i W_i}\sum_{i=1}^{4}\sum_{h=1}^{H_i}\sum_{w=1}^{W_i}(D_i^n(h, w) + D_i^a(h, w)) \quad (2)$$

where $H_i$ and $W_i$ respectively represent the height and width of the features output by $i$-th stage.

### 3.3.2 Abnormality Inverse Mimicking.
Distillation for unsupervised AD generally refers to reducing the distances between the features output by the student and teacher networks on normal images. By default, the student does not learn the feature representation ability of the teacher in anomalous regions. Therefore, the conventional distillation method does not directly train the model on the abnormality features, but only implicitly differentiates the features of the student network and the teacher network in the anomalous regions through forward distillation in the normal regions. However, this kind of forward distillation can easily lead to over-generalization

of abnormality features between student and teacher networks because of the neglect of the training of anomalous regions.

Following the idea of Pull & Push [46], we introduce the distillation method Abnormality Inverse Mimicking, which explicitly gives the meaning of the abnormality features of the student network by maximizing the cosine distances between the features of the student and teacher networks on synthetic anomalies. After remodeling by Abnormality Inverse Mimicking, the student network is able to output abnormality features that are significantly different from the teacher network in anomalous regions, especially anomaly centers.

With the help of anomaly mask and L1 distance, we unify the AIM distillation training of normal images and synthetic anomalous images. Let the input image be $I$, which may be a normal image $I^n$ or an anomalous image $I_a^n$. For each image $I$, let the $i$-th stage output of the teacher network be $F^T_i$, and let the $i$-th stage output of Abnormality Branch of the student network be $F^{S_{AIM}}_i$. Then the AIM distillation loss $L_{NGM}$ is expressed as

$$L_{AIM} = \frac{1}{4}\frac{1}{H_i W_i}\sum_{i=1}^{4}\sum_{h=1}^{H_i}\sum_{w=1}^{W_i}|M_i^{AIM}(h, w) - M_i^{gt}| \quad (3)$$

where $M_i^{gt}$ represents the ground-truth anomaly segmentation mask which is downsampled to the feature size of the $i$-th stage, $M_i^{AIM}$ represents the cosine distance map between teacher and student features of the $i$-th stage, calculated as

$$M_i^{AIM}(h, w) = 1 - \frac{F^T_i(h, w)F^{S_{AIM}}_i(h, w)}{\|F^T_i(h, w)\|\|F^{S_{AIM}}_i(h, w)\|} \quad (4)$$

## 3.4 Multi-perception Segmentation Network
In most previous AD algorithms based on knowledge distillation, such as RD [11] and RD++ [31], the cosine distances between features of each stage of the teacher and student networks are added or multiplied as the final anomaly map. However, since the weights used in fusing the anomaly maps of different stages are the same, the difference in correctness of the anomaly maps is ignored. Besides, due to the structural characteristics of objects in the images, the likelihood of anomaly occurrence varying from region to region. Therefore, anomaly map fusion methods used in DeSTSeg [45] where each pixel in the same anomaly map uses the same weight when fusing do not fully utilize the pattern information of the images, resulting in inaccurate anomaly localization.

To fuse anomaly maps in an optimal way and improve the location accuracy of the fused anomaly map, we propose Multi-perception Segmentation Network in this section. First, we design a data enhancement method Foreground-aware Anomaly Synthesis to control the synthetic anomalies within the foreground of the object images, thereby optimizing the subsequent training and making the fusion network pay more attention to the foreground areas where anomalies are more likely to occur. After that, we propose Multi-perception Segmentation Head to obtain the optimal anomaly map based on multi-perception mechanism.

### 3.4.1 Foreground-aware Anomaly Synthesis.
In recent years, anomaly synthesis methods have been gradually introduced into the design of unsupervised AD algorithms, providing synthetic anomalous images and corresponding binary anomaly masks as the ground truth for the model training process. Among them, the simulated

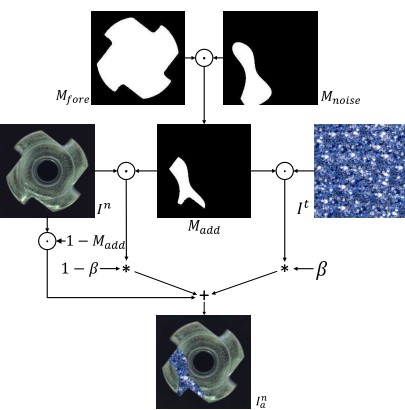

**Figure 5: Foreground-aware Anomaly Synthesis, where $\beta$ and $\odot$ represent opacity and element-wise multiplication.**

anomaly generation method proposed in DRÆM [40] that uses random two-dimensional Perlin noise and the images from the external dataset Describable Textures Dataset (DTD) [8] is widely recognized and introduced into a variety of AD algorithms. However, considering that anomalies in some industrial images may only occur in the foreground where objects are located, synthesizing anomalies based on random full-image noise is likely to result in large differences between synthetic anomalies and real anomalies.

To solve this problem, we make some improvements based on DRÆM and put forward Foreground-aware Anomaly Synthesis. Utilizing the foreground extraction method in traditional image processing field, we realize anomaly synthesis only in the foreground areas of the images, aiming to generate synthetic anomalous images that are closer to real ones. The detailed process is shown in the Fig. 5. First, employ GrabCut [25] algorithm to extract the foreground $M_{fore}$. Then, use binarized Perlin noise $M_{noise}$ restricted to the foreground to superimpose the external texture image as random noise. Finally, add the noise to the normal image $I_n$ and obtain the synthesized anomalous image $I_a^n$.

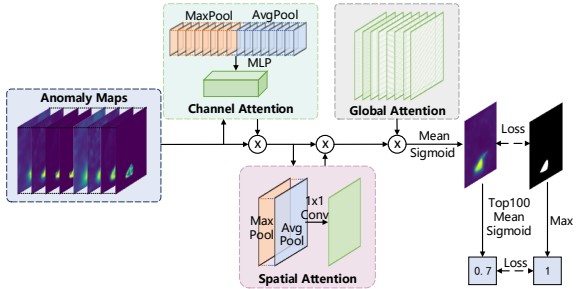

**Figure 6: Multi-perception Mechanism.**

*3.4.2 Multi-perception Segmentation Head.* During the training process of anomaly map fusion, we further freeze the student network. Based on the cosine distances of the extracted features by teachers and students, the preliminary anomaly segmentation results are obtained. Furthermore, the ground-truth anomaly masks

are used to optimize the anomaly segmentation results with the help of multiple perceptions. The process mainly consists of the following three steps.

First, for each test image $I$, a total of eight anomaly maps are calculated based on the pixel-wise cosine similarity using the intermediate features obtained after NGM and AIM distillation in the student network and the features of the teacher network.

Second, based on the ideas mentioned in 3.2, we also introduce **Pyramid Upsampling** to achieve the fusion of high-level and low-level anomaly maps. For the anomaly maps $M_i^{NGM}$, $i = \{1, 2, 3, 4\}$ corresponding to NGM distillation, a top-down fusion path is added. For the abnormal maps $M_i^{AIM}$, $i = \{1, 2, 3, 4\}$ corresponding to AIM distillation, a down-top fusion path is added. For the fused anomaly maps, we uniformly perform bilinear interpolation upsampling operation to expand them to the input image size.

Third, we utilize **Multi-perception Mechanism** to process and fuse the concatenated eight-channel anomaly map, as in Fig. 6. At the start, to enable anomaly segmentation to focus more on stages with stronger anomaly detection capabilities, we introduce a channel attention mechanism based on the eight-channel tensor. Besides, a spatial attention mechanism is added, which effectively makes the anomaly localization process pay more attention to the most likely regions where anomalies occur. Considering that the convolution kernel is unchanged for different regions of the images, we innovatively propose a global attention parameter, to distinguish the weights on different regions of different channels during averaging the processed eight-channel tensor along the channel dimension Finally, after channel compression, pass the tensor through a sigmoid layer to obtain the fused anomaly segmentation map $M$, average the values of the maximum 100 pixels in $M$ with sigmoid to calculate the anomaly score $S$. Here, we optimize segmentation map and the image anomaly score using Binary Cross-Entropy loss (BCE) as

$$L_{seg} = \text{BCE}(M, M^{gt}) + \text{BCE}(S, \max(M^{gt})) \tag{5}$$

## 4 Experiments

### 4.1 Datasets

We conduct relevant experiments on three common unsupervised AD benchmarks. **MVTec AD** [4] is the most widely used unsupervised industrial image AD benchmark for, which contains 5 classes of texture images and 10 classes of object images. 3629 normal images constitute the training set, and 1725 images containing both anomalous and normal images constitute the test set. Similarly, **BTAD** [20] contains 2540 industrial images divided into three categories, where the image partitioning errors have been corrected before the experiments. **MPDD** [17] is a dataset focusing on anomaly detection during painted metal parts fabrication, which contains 6 classes of images. There are 888 normal images in the training set and a total of 458 normal and anomalous images in the test set.

### 4.2 Implementation Details

*Experiment Setting.* Consistent with previous unsupervised AD algorithms, we train corresponding detection models for each category of image in the dataset separately. All images input to the network during the training and inference process are resized to a fixed resolution of $256 \times 256$, and no other image augmentation

**Table 1: Anomaly detection Results I-AUC/P-AUC/PRO (%) on MVTec AD**

| | Category | US [5] | STPM [32] | RD [11] | RD++ [31] | DeSTSeg [45] | MemKD [13] | Pull & Push [46] | Ours |
|---|---|---|---|---|---|---|---|---|---|
| **Textures** | Carpet | 91.6/-/87.9 | -/98.8/95.8 | 98.9/98.9/97.0 | **100**/99.2/97.7 | -/96.1/- | - | 95.9/**99.5**/**98.3** | 99.96/99.26/98.11 |
| | Grid | 81.0/-/95.2 | -/99.0/96.6 | **100**/99.3/97.6 | **100**/99.3/97.7 | -/99.1/- | - | 99.9/99.4/**97.7** | **100**/**99.46**/97.57 |
| | Leather | 88.2/-/94.5 | -/99.3/98.0 | **100**/99.4/99.1 | **100**/99.4/99.2 | -/99.7/- | - | 63.6/99.7/98.7 | **100**/**99.76**/**99.50** |
| | Tile | 99.1/-/94.6 | -/97.4/92.1 | 99.3/95.6/90.6 | 99.7/96.6/92.4 | -/98.0/- | - | 99.7/96.8/90.3 | **100**/**99.53**/**97.43** |
| | Wood | 97.7/-/91.1 | -/97.2/93.6 | 99.2/95.3/90.9 | 99.3/95.8/93.3 | -/97.7/- | - | 99.6/95.2/93.2 | **99.74**/**98.02**/**96.37** |
| | Average | 91.5/-/92.7 | -/98.3/95.2 | 99.5/97.7/95.0 | 99.8/98.1/96.1 | -/98.1/- | - | 91.7/98.1/95.6 | **99.94**/**99.21**/**97.80** |
| **Objects** | Bottle | 99.0/-/93.1 | -/98.8/95.1 | **100**/98.7/96.6 | **100**/98.8/97.0 | -/99.2/- | - | 99.9/98.7/95.6 | **100**/98.99/**97.07** |
| | Cable | 86.2/-/81.8 | -/95.5/87.7 | 95.0/97.4/91.0 | **99.2**/**98.4**/**93.9** | -/97.3/- | - | 98.4/95.7/87.5 | 98.43/97.97/92.10 |
| | Capsule | 86.1/-/96.8 | -/98.3/92.2 | 96.3/98.7/95.8 | 99.0/98.8/96.4 | -/**99.1**/- | - | **99.8**/97.8/89.9 | 99.60/99.01/**97.06** |
| | Hazelnut | 93.1/-/96.5 | -/98.5/94.3 | 99.9/98.9/95.5 | **100**/99.2/96.3 | -/99.6/- | - | 99.5/98.6/96.1 | 99.93/**99.28**/**96.92** |
| | Metal_nut | 82.0/-/94.2 | -/97.6/94.5 | **100**/97.3/92.3 | **100**/98.1/93.0 | -/98.6/- | - | 86.9/97.8/93.2 | 99.90/**98.40**/**95.25** |
| | Pill | 87.9/-/96.1 | -/97.8/96.5 | 96.6/98.2/96.4 | 98.4/98.3/97.0 | -/98.7/- | - | **99.7**/98.6/94.9 | 98.15/**99.33**/**97.77** |
| | Screw | 54.9/-/94.2 | -/98.3/93.0 | 97.0/99.6/98.2 | 98.9/**99.7**/**98.6** | -/98.5/- | - | 85.3/96.9/85.6 | 96.19/99.32/97.32 |
| | Toothbrush | 95.3/-/93.3 | -/98.9/92.2 | 99.5/99.1/94.5 | **100**/99.1/94.2 | -/99.3/- | - | 94.0/99.1/91.8 | **100**/**99.41**/95.81 |
| | Transistor | 81.8/-/66.6 | -/82.5/69.5 | 96.7/92.5/78.0 | 98.5/94.3/81.8 | -/89.1/- | - | **100**/**99.2**/**97.6** | 99.17/95.63/85.87 |
| | Zipper | 91.9/-/95.1 | -98.5/95.2 | 98.5/98.2/95.4 | 98.6/98.8/96.3 | -/99.1/- | - | 99.6/97.9/93.0 | **99.87**/**99.41**/**97.77** |
| | Average | 85.8/-/90.8 | -/96.5/90.9 | 98.0/97.9/93.4 | **99.2**/98.4/94.5 | -/97.9/- | - | 96.3/98.0/92.5 | 98.94/**98.49**/**95.10** |
| | Total Average | 87.7/-/91.4 | 95.5/97.0/92.1 | 98.5/97.8/93.9 | 99.4/98.3/95.0 | 98.6/97.9/- | **99.6**/98.2/94.5 | 94.8/98.1/93.6 | 99.40/**98.85**/**96.13** |

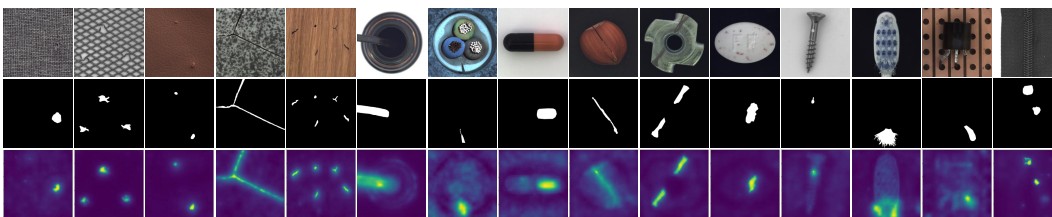

**Figure 7: Visualization example of DMDD's anomaly detection results of 15 classes of images, showing anomalous images, ground-truth masks and anomaly maps (the yellower the regions, the higher the probability of anomalies) from top to bottom.**

**Table 2: Anomaly detection Results P-AUC/PRO (%) on BTAD**

| Category | FastFlow [38] | PatchCore [24] | RD [11] | RD++ [31] | Ours |
|---|---|---|---|---|---|
| Class 01 | **97.1**/71.7 | 97.03/64.92 | 96.6/75.3 | 96.2/73.2 | 96.95/**79.92** |
| Class 02 | 93.6/63.1 | 95.83/47.27 | 96.7/68.2 | 96.4/71.3 | **97.33**/74.25 |
| Class 03 | 98.3/79.5 | 99.19/67.72 | 99.7/**87.8** | 99.7/87.4 | **99.89**/86.67 |
| Average | 96.33/71.43 | 97.35/59.97 | 97.67/77.10 | 97.43/77.30 | **98.06**/**80.28** |

**Table 3: Anomaly detection Results I-AUC/P-AUC/PRO (%) on MPDD**

| Category | RD [11] | RD++ [31] | MemKD [13] | Ours |
|---|---|---|---|---|
| bracket black | 90.36/98.14/92.35 | 89.10/98.05/92.59 | - | **95.74**/**98.57**/**97.18** |
| bracket brown | 92.76/97.08/95.11 | 95.02/97.15/94.58 | - | **98.94**/**97.31**/**96.11** |
| bracket white | 87.44/99.32/97.65 | 90.11/99.43/97.25 | - | **98.33**/**99.66**/**99.29** |
| connector | **100**/99.45/96.90 | **100**/99.29/95.87 | - | **100**/**99.68**/**98.28** |
| metal plate | **100**/**99.09**/96.09 | **100**/99.08/96.15 | - | **100**/99.08/**96.51** |
| tubes | **96.06**/99.12/97.23 | 94.52/99.13/97.39 | - | 95.56/**99.44**/**98.58** |
| Average | 94.44/98.70/95.89 | 94.79/98.69/95.64 | 95.4/98.4/95.9 | **98.10**/**98.96**/**97.66** |

methods are used except for anomaly synthesis. During the training process, the student network is optimized by Adam optimizer for 100-400 epochs with a learning rate of 0.005. The experiments are all completed based on PyTorch on a single Nvidia GTX 3090 GPU.

*Evaluation Metrics.* We report area under the ROC curve (AUC) as the evaluation metrics for image-level detection and pixel-level localization, abbreviated as I-AUC and P-AUC. For anomaly localization, per-region-overlap (PRO) [5] is also used for comparison.

### 4.3  Main Results

*Anomaly Detection on MVTec AD.* Table 1 reports the anomaly detection and localization results of the advanced KD-based unsupervised AD methods and our proposed DMDD on MVTec AD.

The average pixel-level anomaly localization results of DMDD are proved to outperform other KD methods to reach SOTA. Notably, our method surpasses the previous methods in all metrics on texture images, and exceeds RD++ [31] which is the current KD-based SOTA method for unsupervised AD task by 0.55% and 1.13% on total average P-AUC and PRO. The anomaly detection qualitative results are visualized in Fig. 7.

*Anomaly Detection on BTAD.* We exhibit the experimental results of DMDD on BTAD as in Table 2. Compared with previous unsupervised AD methods, DMDD achieves the best average localization performance, reaching 98.06% and 80.28% on P-AUC and PRO.

**Table 4: Ablation Study on Student-Teacher Network Architecture.**

| PMN (Inner) | PMN (Outer) | I-AUC | P-AUC | P-PRO |
|---|---|---|---|---|
| Normality Branch With NGM Distillation | | | | |
| - | - | 95.76 | 97.72 | 93.89 |
| ✓ | - | 98.66 | 98.19 | 94.90 |
| ✓ | ✓ | 98.59 | 98.31 | 94.91 |
| Abnormality Branch With AIM Distillation | | | | |
| - | - | 98.16 | 98.17 | 94.10 |
| ✓ | - | 98.72 | 98.51 | 94.93 |
| ✓ | ✓ | 99.24 | 98.77 | 95.38 |
| Decouple S-T Network With DMD | | | | |
| ✓ | ✓ | 99.18 | 98.72 | 95.70 |

*Anomaly Detection on MPDD.* We perform experimental validation of anomaly localization on MPDD, and the related results are reported in Table 3. To better compare with the KD-based methods, we re-conduct the experiments related to RD [11] and RD++ [31] on MPDD and record the results exactly according to the original settings of these two methods. Obviously, our method is more outstanding in anomaly detection and localization capability.

## 4.4 Ablation Studies

*Ablation Study on Student-Teacher Network Architecture.* We first conduct ablation experiments on the structure of Decoupled Student-Teacher (S-T) Network to investigate the necessity of the dual-branch design and the effectiveness of the proposed Pyramid Modeling Network (PMN). Table 4 is divided into three parts from top to bottom, showing the anomaly detection performance of only Normality Branch with NGM distillation, only Abnormality Branch with AIM distillation, and the whole Decoupled Student-Teacher Network with Dual-Modeling Distillation (DMD). The significant improvement of PRO with the similar AUCs proves the localization superiority of the dual-branch design over the single branch. In addition, we experimentally evaluate the inclusion of PMN, and it can be seen from Table 4 that the anomaly detection effect is significantly improved by simultaneously adding two multi-scale fusion paths, inner and outer, of PMN.

Some relevant visualization results are exhibited in Fig. 8, mainly including the anomaly maps of each stage output by the two branches with or without PMN. It is evident that the detection of anomaly maps of all stages is improved by the addition of PMN, which proves that PMN is capable of increasing the accuracy of anomaly localization. In addition, it is intuitively clear from Fig. 8 that Normality Branch and Abnormality Branch pay attention to different regions of anomalies. Compared to Normality Branch, Abnormality Branch only focuses on the center region of anomalies, which explains why the dual-branch design yields the best detection performance.

*Ablation Study on Multi-perception Segmentation Network Components.* As shown in Table 5, compared to anomaly map fusion using only plain summation (w/o MSN), our proposed Multi-perception

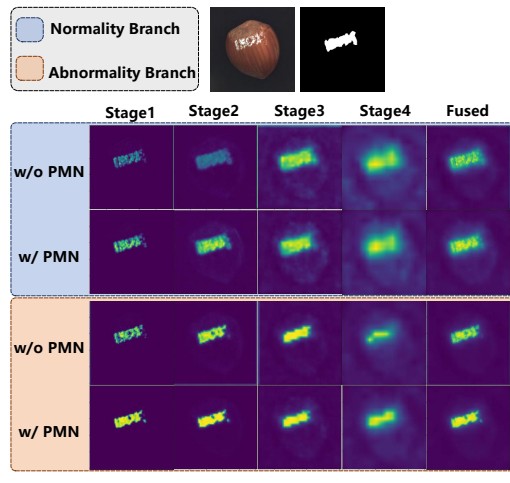

**Figure 8: Visualization of anomaly maps associated with ablation study on Student-Teacher Network architecture.**

**Table 5: Ablation Study on Multi-perception Segmentation Network Components.**

| | I-AUC | P-AUC | P-PRO |
|---|---|---|---|
| w/o MSN | 99.18 | 98.72 | 95.70 |
| w/ MSN | 99.40 | 98.85 | 96.13 |
| -FAS | 99.16 | 98.78 | 95.95 |
| -PU | 99.25 | 98.60 | 95.39 |
| -MM | 98.61 | 98.72 | 95.77 |

Segmentation Network (MSN) improves I-AUC, P-AUC and PRO by 0.22%, 0.13% and 0.43% respectively. In addition, we conduct ablation studies on the components of Multi-perception Fusion Network. It turns out that the removal of Foreground-aware Anomaly Synthesis (FAS), i.e.,using full-image anomaly synthesis, or the subtraction of Pyramid Upsampling (PU) or Multi-perception Mechanism (MM) in Multi-perception Segmentation Head, all make DMDD less effective at detecting and localizing anomalies, which squarely justifies the need for these modules.

## 5 Conclusion

In this paper, we propose a novel unsupervised anomaly detection method based on knowledge distillation paradigm DMDD. A Decoupled Student-Teacher network is employed to differentiate the network structures while ensuring the same information capacity for the teacher and student networks. Dual-Modeling Distillation is proposed to distill student features from both normality simulation and abnormality distancing, allowing the anomaly maps derived from the feature similarity of student and teacher to focus on both the anomaly edge and center. Besides, to optimize the anomaly segmentation process, Multi-perception Segmentation Network is presented. Experimental outcomes demonstrate that comparing with the previous knowledge distillation methods, our method significantly improves the image anomaly localization effect.

# Acknowledgments

This work is supported by National Natural Science Foundation of China 62372032 and National Key Research and Development Program of China under Grant 2022YFC3320402.

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
