# OpenReview forum: "Dual-Modeling Decouple Distillation for Unsupervised Anomaly Detection"
_acmmm.org/ACMMM/2024/Conference — MM2024 Poster_

### Official Review · Reviewer_EN35 · 2024-05-02

**Rating:** 5
**Confidence:** 3

**Summary:**

This paper proposes a novel Dual-Modeling Decouple Distillation for the unsupervised anomaly detection. In DMDD, a Decouple Student Network is proposed to decouple the initial student features into normality and abnormality features. The decoupled features are then distilled through Normality Guidance Modeling and Abnormality Inverse Mimicking to highlight their normality and abnormality.

**Strengths:**

Decoupling features of the student network into normality feature and abnormality feature is interesting. The experimental results show the proposed method can achieve SOTA anomaly localization results on popular AD datasets.

**Limitations:**

[Major]
1. In line 344-347, “feature of anomalous region to be decomposed into the normality feature and abnormality feature”. Decoupling feature into normality and abnormality feature is the core idea in your paper. But, what do normality and abnormality represent respectively? Why can a feature be decomposed into normality and abnormality features? This is not well explained in the paper, a more intuitive explanation is needed.

2. During training, does the method have to have anomalies or pseudo anomalies? If only with normal samples, is it still usable?

3. In Figure 6, it would be better to compare with other SOTA methods, while not just provided the qualitative results of your method.

4. Implementing the proposed method is non-trivial, will the code be open source? As a good work, open source code is required to ensure reproducibility.

[Minor]

1.In line 226 and 229, the $I_1, I_2$, … needs to be wrapped by {…}. In line 504, there should be $L_{AIM}$. In figure 2, the right part feature map F is hard to see clearly.

**Suitability:**

2

---

### Official Review · Reviewer_9CZG · 2024-05-08

**Rating:** 2
**Confidence:** 3

**Summary:**

This paper presents a novel unsupervised image anomaly detection method called DMDD. Student-Teacher network is employed to decouple student features into normality and abnormality features. Dual-Modeling Distillation is used to distill student features. Multi-perception Segmentation Network is used to optimize the anomaly segmentation process. Experiments on three AD datasets demonstrate its performance over existing approaches.

**Strengths:**

Unsupervised image anomaly detection is an interesting research topic with important applications in various fields. The strengths of this paper are:

1. Proposing a new model based on knowledge distillation to solve the problem.
2. The discussion of difference between edge and center for normal and abnormal samples is interesting, but needs justification.

**Limitations:**

There are several key drawbacks for this paper:

1. The writing needs to be improved to make it easier to understand and more logically sound. For example, the introduction section is hard to follow when it comes to the authors' method design. It is suggested that the authors rephrase it to make it clearer.
2. S3.2, "Existing methods only use one of the above methods, which leads to insufficient perception of the anomalies’ center or edge, resulting in inaccurate anomaly localization results." What does the anomalies' center and edge mean? How does it affect the anomaly detection process? The statements needs clarification and experiments to prove this point.
3. The author names the proposed method as an unsupervised one. However, the training used synthesized abnormal samples by stacking normal samples and noise. The question is, if you have no knowledge about how actual abnormal samples look like, how can you generate good synthetic abnormal samples? Are all the methods being compared also used synthesized abnormal samples for training? If no, then the comparison is unfair.
4. The citation is insufficient, e.g., GrabCut (line 589, left column)

A few typos:

1. Line 84, right column, "KD-based" instead of "KD-base". There are also a few other "KD-base" in the paper, which should be modified.
2. Line 84, right column, "assume" instead of "assumpt"
3. Line 89, right column, "in practice" instead of "in practical"

**Suitability:**

2

---

### Official Review · Reviewer_zDNs · 2024-05-23

**Rating:** 3
**Confidence:** 3

**Summary:**

This paper proposes a Dual-Modeling Decouple Distillation method (DMDD) for the unsupervised Anomaly Detection. The experimental results demonstrate the effectiveness of the author's method. However, the experimental results seem to be incomplete. Representative comparison methods (PatchCore [24]) were not run on all datasets.

**Strengths:**

1.The experimental results demonstrate that the performance of DMDD exceeds previous state-of-the-art (SOTA) methods based on knowledge distillation.

2.The author employs a dual-branch dual-modeling decoupling distillation approach to perform distinct distillation.

**Limitations:**

1.The comparison with the key method, PatchCore [24], should encompass all datasets in order to enhance the persuasiveness of the proposed method.

2.This paper may benefit from outlining the key contributions in the introduction in a concise manner, in order to help readers grasp the differences between the authors' method and other existing approaches quickly.

3.The MemKD column in Table 3 is empty, yet there is an average result for it. Additionally, the textual description does not mention MemKD. The same situation occurs in Table 1.The author should provide complete results.

4.The author should emphasize the distinctiveness of the proposed method in comparison to existing methods, as the current discussion lacks this aspect.

5.The author has not released the code, lacking reproducibility.

6.What is “improving distillation content from normality or abnormality only has the limitation of anomaly focus”? Is it referring to the limitation caused by favoring one aspect too much?

**Suitability:**

2

---

### Official Review · Reviewer_sqi4 · 2024-05-24

**Rating:** 5
**Confidence:** 3

**Summary:**

The paper 'Dual-Modeling Decouple Distillation for Unsupervised Anomaly Detection' proposes an interesting approach for AD in images, based on reverse distillation. It is a well written paper, with minor details that need correction.
-  The paper introduces a new approach, Dual-Modeling Decouple Distillation (DMDD). It decouples the student & teacher networks for abnormality and normality learning.
- The paper presents a robust set of experiments, and ablation studies. The proposed approach outperforms the SoTa, with multiple metrics.

**Strengths:**

1. Decoupled Student Teacher Network: The underlying ideation and motivation is crisply explained (Sec 3.2).
2. Pyramid Modeling Network : design is well thought out. The motivation of the abnormatity and the normality branch makes sense. The overall idea of receptive field behaving differently, for normal pixels vs anomalous pixels is correct.
3. Dual-Modeling Distillation: the loss function, and the overall architecture conceptually differentiates mining the anomalies from normal samples.
4. the set of experiments are exhaustive and comprehensive. Its good that the paper visits more than one metric.
5. the ablation study provided is useful

**Limitations:**

1.Minor errors:
- Line 84 : KD-base methods assumpt -> assume or make assumption
- Line 344: features , not feature. It is a collection of features, not a single one.
-
2. The core idea discussed in lines 370 - 380 lacks a clarity
3. the apper is overall well organized, but clarity in some parttscould be improved. Long complex sentences are a hindrance to comprehension.
4. the conclusion section could have been informative, with a bit further light on future work.
5. Lines: 95-98 : This is a  strong assumption. A paper should present a hypothesis,  not a strong assumption as a premise.  Introduction section should avoid as such.

**Suitability:**

3

---

### Meta-Review · Area_Chair_17Ys · 2024-07-02

**Recommendation:** Accept (Poster)
**Confidence:** 4

**Metareview:**

This paper originally received scores of 2 Weak Accept, 1 Borderline Reject, and 1 Weak Reject.

According to the reviewers and my evaluation, the paper has clear strengths in its interesting and well-designed model and its good performance. However, the issues and limitations are mainly related to the representations of some claims and discussions. These need to be addressed in the final version according to the reviews.